

# A GAN-based approach to solar radiation prediction: data augmentation and model optimization for Saudi Arabia

Abdalla Alameen and Sultan Mesfer Aldossary

Department of Computer Engineering and Information, Prince Sattam Bin Abdulaziz University, Wadi ad-Dawasir, Riyadh, Saudi Arabia

## ABSTRACT

**Background:** Accurate solar radiation prediction is essential for optimizing renewable energy systems but remains challenging due to data scarcity and variability. This study addresses these challenges by employing generative adversarial networks (GANs) to generate high-quality synthetic solar radiation data.

**Methods:** A novel framework was developed that integrates GAN-generated synthetic data with machine learning and deep learning models, including CNN-LSTM architectures. These models were trained and evaluated using augmented datasets to improve predictive accuracy and adaptability across diverse climatic zones.

**Results:** Models trained on augmented datasets exhibited significant improvements, with root mean square error (RMSE) reduced by 15.2% and mean absolute error (MAE) decreased by 19.9%. The framework effectively bridged data gaps and enhanced model generalization, enabling applicability across various climatic regions in Saudi Arabia.

**Conclusions:** The proposed framework facilitates practical applications such as photovoltaic system optimization, grid stability enhancement, and resource planning. By aligning with Saudi Arabia's Vision 2030 and global renewable energy objectives, this study presents a scalable and adaptable approach to advancing renewable energy systems. However, challenges such as computational complexity and hyperparameter sensitivity warrant further investigation, providing a robust pathway toward sustainable energy futures worldwide.

## INTRODUCTION

The development of Saudi Arabia's energy landscape is closely tied to the nation's progress in technological innovation, economic diversification, and social well-being (*Al-Gahtani, 2024*). For decades, the Saudi economy has been heavily dependent on vast reserves of non-renewable fossil fuels- primarily oil and natural gas- which have fueled industrial growth, transportation, and household energy consumption both domestically and internationally (*Islam & Ali, 2024*). However, the finite nature of these resources presents a growing challenge amid rising global population, accelerated industrialization, and

Corresponding author
Abdalla Alameen,
a.alameen@psau.edu.sa

increasing energy demands. Continued reliance on fossil fuels raises sustainability concerns related to resource depletion and the environmental impact of carbon emissions (*Al-Gahtani, 2024*).

In response, Saudi Arabia has positioned renewable energy as a strategic priority within its Vision 2030 initiative (*Kingdom of Saudi Arabia, 2016*). This national agenda aims to diversify the economy, reduce dependence on oil, and establish the Kingdom as a global leader in sustainable energy development (*Wang & Azam, 2024*). Solar energy, in particular, plays a central role due to the country's abundant solar resources and favorable geographic conditions (*Ahmed et al., 2020*). Major initiatives such as the Sakaka PV Plant and the NEOM City Solar Project depend on accurate solar radiation forecasts for effective planning, design, and operations (*Islam & Ali, 2024*). Leveraging this solar potential is not only economically advantageous but also essential for environmental conservation and aligns with international efforts to mitigate climate change (*Zabelin, 2024*).

Despite its promise, the integration of solar energy into the national energy mix requires accurate and scalable solar radiation forecasting models (*Husainy et al., 2024*). Traditional methods, such as numerical weather prediction (NWP) and statistical models, provide foundational tools. However, they suffer from significant limitations, such as high computational demands, reliance on high-quality input data, and an inability to effectively capture the nonlinear and dynamic nature of solar radiation (*Krishnan, Kumar & Inda, 2023*). Furthermore, these models often lack adaptability to regional climatic variations—an important consideration in geographically diverse countries like Saudi Arabia.

Recent advances in machine learning (ML) and deep learning (DL) have addressed some of these limitations, offering enhanced capabilities for capturing complex data patterns. Nevertheless, these approaches typically require large volumes of high-resolution data, which are often unavailable in areas with limited meteorological infrastructure, such as the Kingdom's desert, coastal, and mountainous regions (*Akkem, Biswas & Varanasi, 2024*). This limitation underscores the need for innovative methods capable of generating high-quality synthetic data to supplement existing datasets and enhance model performance.

Generative adversarial networks (GANs) offer a transformative solution to the challenge of data scarcity. GANs generate high-quality synthetic data that mimics real-world solar radiation patterns, enabling the augmentation of limited datasets and enhancing the robustness and generalization capabilities of predictive models (*Nematchoua, Orosa & Afaifia, 2022*).

GANs were selected over other data augmentation techniques due to their unique ability to produce diverse and realistic synthetic data (*Goodfellow et al., 2020*). GANs, unlike traditional methods like oversampling or interpolation, use adversarial training between a generator and a discriminator, resulting in data that captures complex distributions and variability (*Figueira & Vaz, 2022*). This capability is especially crucial for modeling solar radiation in diverse climatic zones such as Riyadh, Jeddah, Abha, and Taif, where distinct desert, coastal, and mountainous features influence radiation patterns. Furthermore, extensions like conditional GANs (cGANs) enable the incorporation of meteorological and

environmental variables, allowing synthetic data to be tailored to specific regional conditions (*Saxena & Cao, 2021*).

Despite advances in solar radiation prediction, the existing literature largely emphasizes conventional machine learning and deep learning models, which rely heavily on extensive real-world datasets (*Liu et al., 2022*). These studies often overlook the potential of GANs to augment data-scarce environments (*Kumari & Toshniwal, 2021*).

This study addresses the identified gap by proposing a GAN-based framework that generates high-fidelity synthetic solar radiation data and integrates it with advanced predictive models, such as CNN-LSTM, to enhance forecast accuracy and scalability. By targeting the diverse climatic zones of Saudi Arabia, the proposed approach offers a robust solution that aligns with the Kingdom's Vision 2030 objectives and contributes to broader global efforts to advance renewable energy systems.

Although GANs have been widely applied for data augmentation in domains such as medical imaging (*Figueira & Vaz, 2022*), their application in solar radiation prediction—particularly in capturing region-specific climatic variability—remains underexplored. In contrast to methods such as variational autoencoders (VAEs) (*Kaur, Islam & Mahmud, 2021*) and diffusion models (*Hatanaka et al., 2023*), generative adversarial networks (GANs) uniquely preserve spatiotemporal correlations in solar radiation data through adversarial training, as evidenced by the Wasserstein distance metrics presented in 'Results'.

This article is organized as follows. 'State of the Art' reviews recent advances in solar radiation prediction, with an emphasis on methodologies suited for data-scarce regions, and identifies gaps in region-specific modeling. 'Theoretical Bases' establishes the theoretical foundations of GANs and their applicability to the synthesis of solar data. 'Materials and Methods' describes the proposed GAN-based framework, its integration with predictive models, and the underlying computational infrastructure. 'Reproducibility' details the experimental setup and the reproducibility measures. 'Results' presents an evaluation of the performance of the framework and its contributions to solar energy forecasting. 'Discussion' interprets the results in the context of the existing literature and discusses limitations. 'Future Directions' outlines the broader implications and future research directions.

## STATE OF THE ART

Over the past decades, methodologies for solar radiation prediction have evolved significantly, transitioning from traditional statistical models to advanced machine learning (ML) and deep learning (DL) approaches. Recently, GANs have emerged as a transformative solution, addressing challenges such as data scarcity and enhancing prediction accuracy through data augmentation. This section reviews these methodologies, focusing on their contributions, limitations, and relevance to solar radiation forecasting in Saudi Arabia, in alignment with Vision 2030 goals (*Kingdom of Saudi Arabia, 2016*).

## Traditional approaches
### Statistical models

Statistical models rely on historical meteorological data to establish relationships between input variables and solar radiation. For example, *Rehman & Ghori (2000)* employed geostatistical techniques to estimate solar radiation variability across Saudi Arabia. Similarly, *Khalil & Rahoma (2021)* developed a cubic empirical model to predict diffuse solar radiation in Jeddah, achieving high precision. However, these models often struggle to capture nonlinear dependencies and depend heavily on reliable long-term datasets, limiting their adaptability for dynamic and real-time forecasting scenarios (*Waheed et al., 2024*).

### Physical models

Physical models simulate atmospheric processes—such as cloud cover, aerosol concentration, and radiative transfer—to estimate solar radiation. *Alnaser, Trieb & Knies (2007)* utilized climatic parameters to forecast solar radiation across the Arabian Peninsula. *Farahat, Kambezidis & Labban (2023)* analyzed meteorological data to investigate spatial and temporal variability within Saudi Arabia. Although physical models offer valuable climatological insights, their high computational demands and sensitivity to atmospheric uncertainties diminish their applicability for real-time forecasting (*Xu et al., 2021*).

## Machine learning approaches

Machine learning techniques, including random forest and gradient boosting, have shown notable improvements in solar radiation prediction. *Chaibi et al. (2018)* combined satellite and ground-based observations using these models in Riyadh, enhancing accuracy compared to models trained on standalone datasets. However, ML models often require region-specific retraining and are computationally intensive, which limits their scalability.

## Deep learning approaches

Deep learning models, such as artificial neural networks (ANNs) and convolutional neural networks (CNNs), further advance prediction capabilities by learning complex nonlinear relationships. *Hanif et al. (2024)*, for instance, incorporated meteorological variables like wind speed and humidity into ANN models in Dammam, significantly improving daily predictions. CNNs are proficient at extracting spatial features, while long short-term memory (LSTM) networks excel at modeling temporal dependencies (*Song et al., 2020*). Despite their strong performance, DL models require large volumes of high-quality data and careful preprocessing to prevent overfitting.

## Hybrid models

Hybrid models, which integrate physical, statistical, and ML approaches, have also gained prominence. *Hedar et al. (2021)* combined weather simulation outputs with ML models to improve solar radiation forecasts in eastern Saudi Arabia, demonstrating enhanced performance under extreme conditions. Similarly, *Khan & Khan (2024)* implemented a CNN-LSTM hybrid model in the UAE to capture spatiotemporal dependencies, with

promising results applicable to the Saudi context. Nevertheless, these hybrid approaches are computationally expensive and often rely on extensive data infrastructure, posing challenges for real-time implementation in data-scarce regions.

Emerging hybrid frameworks (*Delarami, Mohammadbeigi & Gharib, 2024*) and edge-computing optimizations (*Minh et al., 2022*) have further improved the feasibility of generative data augmentation for renewable energy systems, supporting the scalable deployment goals outlined in 'Future Directions'.

### GAN-based approaches

GANs, introduced by *Goodfellow et al. (2020)*, comprise two neural networks—a generator and a discriminator—that compete in an adversarial process. The generator synthesizes data, while the discriminator assesses its authenticity. This dynamic training process enables GANs to produce realistic synthetic data, effectively augmenting training datasets and improving model performance (*Fayaz et al., 2024*).

Recent advancements in GAN architectures, such as StyleGAN and BigGAN, have expanded their capabilities. StyleGAN enables fine-grained control over generated outputs, allowing synthetic solar radiation data to be customized for specific regional climates (*Kousounadis-Knousen et al., 2023*). BigGAN, on the other hand, enhances scalability and fidelity through the use of larger batch sizes and advanced normalization strategies (*Quaicoo et al., 2024*). These innovations are especially promising for improving prediction accuracy and adaptability in data-limited settings like Saudi Arabia.

### Summary

The evolution of solar radiation prediction methodologies reflects a clear transition from traditional statistical and physical models to more advanced ML, DL, and GAN-based techniques. While early models provide valuable foundational understanding, their limitations in handling nonlinearities and adapting to real-time environments necessitate the adoption of modern approaches. ML and DL models offer superior spatial-temporal modeling capabilities but require extensive data and computational resources. GANs emerge as a compelling alternative, addressing data scarcity while improving predictive accuracy through realistic data augmentation. Future research should prioritize the adaptation of GAN architectures to Saudi Arabia's diverse climatic regions and the integration of these models with real-time data pipelines to maximize their practical impact on solar radiation forecasting.

## THEORETICAL BASES

The theoretical foundations of this study provide a robust framework for addressing the challenges of solar radiation prediction, particularly in data-scarce regions such as Saudi Arabia. These foundational concepts are essential for constructing a reliable and scalable model capable of capturing the complex, nonlinear dynamics of solar radiation. They underpin the development of an advanced framework that combines data augmentation using GANs with predictive modeling techniques.

## Solar radiation modeling

Solar radiation modeling is fundamental to understanding the atmospheric and environmental factors that influence solar energy availability. A robust modeling framework ensures that the synthetic data generated by the proposed system aligns with real-world patterns and captures the inherent variability of solar radiation (*Izzi, Martella & Longo, 2024*).

The modeling process begins with the calculation of global horizontal irradiance (GHI), which is derived from three components: direct normal irradiance (DNI), diffuse horizontal irradiance (DHI), and the cosine of the solar zenith angle. This relationship is expressed as:

$$GHI = DNI \cdot \cos(\theta) + DHI. \tag{1}$$

This equation serves as the foundation for simulating the direct, diffuse, and angular components of solar radiation—an essential element for photovoltaic (PV) system optimization.

To enhance this baseline approach, radiative transfer models are employed. These models simulate the interaction between solar radiation and atmospheric elements such as aerosols, clouds, and gases (*Zhang et al., 2024*). By incorporating these physical processes, radiative transfer models ensure that the synthetic data accurately reflects region-specific solar radiation characteristics—particularly vital in the context of Saudi Arabia's varied climatic zones.

However, physical models alone often fall short in capturing local variability. To address this, empirical corrections are introduced, adjusting model outputs to account for unique climatic conditions in specific regions—such as the desert and coastal areas of the Kingdom (*Daxini, 2024*). These corrections bring the synthetic data into closer alignment with observed solar radiation patterns, enabling more effective generalization across regions with distinct meteorological profiles.

## Machine learning and deep learning fundamentals

The complex and nonlinear nature of solar radiation data necessitates the application of advanced ML and DL techniques. These methods are capable of uncovering intricate patterns, enabling accurate forecasting even in climatically diverse regions (*Bamisile et al., 2022*).

At the core of ML approaches lies ensemble learning, which combines multiple base learners to enhance predictive performance (*Alam et al., 2023*). This process is mathematically represented as:

$$\hat{y} = \sum_{i=1}^{n} w_i \cdot h_i(x) \tag{2}$$

where $\hat{y}$ is the final predicted output, $w_i$ is the weight assigned to the $i$-th model, and $h_i(x)$ represents the prediction from the $i$-th model. Augmented datasets significantly improve ensemble learning by providing the diverse and high-quality data needed for more accurate and reliable predictions (*Bamisile et al., 2022*).

DL models like CNNs and LSTM networks offer additional capabilities. Training of these networks is typically achieved *via* backpropagation, which iteratively updates model weights to minimize prediction error (*Yahiaoui & Assas, 2023*):

$$w_{ij} = w_{ij} - \eta \cdot \frac{\partial L}{\partial w_{ij}} \tag{3}$$

where $w_{ij}$ is the weight between neurons $i$ and $j$, $\eta$ is the learning rate, and $\frac{\partial L}{\partial w_{ij}}$ is the gradient of the loss function with respect to the weight.

## Generative adversarial networks

GANs have revolutionized data generation by addressing scarcity and improving the robustness of predictive models. Through adversarial training, GANs generate synthetic data that closely resembles real-world observations (*Wen et al., 2023*).

At the heart of a GAN are two neural networks—the generator ($G$) and the discriminator ($D$)—trained simultaneously in a competitive setting (*Nematchoua, Orosa & Afaifia, 2022*). This adversarial setup is captured in the GAN's minimax objective function:

$$\min_{G} \max_{D} V(D, G) = \mathbb{E}_{x \sim p_{data}(x)}[\log D(x)] + \mathbb{E}_{z \sim p_z(z)}[\log(1 - D(G(z)))]. \tag{4}$$

Conditional GANs (cGANs) introduce additional flexibility by incorporating conditional inputs $y$ (*Saxena & Cao, 2021*):

$$\min_{G} \max_{D} V(D, G) = \mathbb{E}_{x,y \sim p_{data}(x|y)}[\log D(x|y)] + \mathbb{E}_{z \sim p_z(z), y \sim p_{data}(y)}[\log(1 - D(G(z|y)|y))]. \tag{5}$$

To overcome challenges like mode collapse, Wasserstein GANs (WGANs) are employed, using the Wasserstein distance metric (*Khare, Wadhvani & Shukla, 2022*):

$$W(p_r, p_g) = \sup_{||f||_L \leq 1} \mathbb{E}_{x \sim p_r}[f(x)] - \mathbb{E}_{x \sim p_g}[f(x)]. \tag{6}$$

## Evaluation metrics

Robust evaluation metrics are essential for validating both the quality of synthetic data and the performance of predictive models. The root mean square error (RMSE) is defined as *Mustafa et al. (2022)*:

$$\text{RMSE} = \sqrt{\frac{1}{n} \sum_{i=1}^{n} (\hat{y}_i - y_i)^2}. \tag{7}$$

The mean absolute error (MAE) offers an intuitive measure of prediction errors (*Boubaker et al., 2021*):

$$\text{MAE} = \frac{1}{n} \sum_{i=1}^{n} |y_i - \hat{y}_i|. \tag{8}$$

The structural similarity index (SSIM) evaluates the similarity between real and synthetic data (*Boubaker et al., 2021*):

$$\text{SSIM}(x, y) = \frac{(2\mu_x\mu_y + C_1)(2\sigma_{xy} + C_2)}{(\mu_x^2 + \mu_y^2 + C_1)(\sigma_x^2 + \sigma_y^2 + C_2)}. \tag{9}$$

The Kullback-Leibler (KL) Divergence measures distribution alignment (*Dairi, Harrou & Sun, 2021*):

$$D_{\text{KL}}(P||Q) = \sum_i P(i) \log \frac{P(i)}{Q(i)}. \tag{10}$$

By integrating these theoretical components—ranging from physical and statistical modeling to GAN-based data augmentation and advanced DL architectures—this framework addresses the core challenges of solar radiation forecasting in data-scarce regions. The proposed approach not only ensures high model performance but also aligns with the sustainable energy goals of Saudi Arabia's Vision 2030.

## MATERIALS AND METHODS

### Overview

The methodology for this study, illustrated in Fig. 1, outlines a comprehensive framework designed to address data scarcity in solar radiation prediction across Saudi Arabia. As shown in the figure, our approach integrates advanced data generation techniques with state-of-the-art predictive modeling to ensure accurate and scalable forecasts tailored to the Kingdom's diverse climatic regions, including desert, coastal, and mountainous areas.

### *Data sources*

The study integrates satellite-based observations, ground-truth solar measurements, and meteorological variables to ensure comprehensive modeling of solar radiation variability across Saudi Arabia:

- **Satellite imagery:** Level 1.5 data from the Spinning Enhanced Visible and Infrared Imager (SEVIRI) sensor aboard the Meteosat Second Generation (MSG) satellites were acquired from the EUMETSAT Data Store. The dataset spans multiple years and includes radiance channels relevant to solar radiation modeling. Satellite imagery was preprocessed and clipped to the Saudi Arabian region using open-source tools, including Satpy and SEVIRI-Reader.
- **Ground-based measurements:** Meteorological stations in Riyadh, Jeddah, Dhahran, and Abha deliver accurate point-source measurements of global horizontal irradiance (GHI), direct normal irradiance (DNI), and diffuse horizontal irradiance (DHI), serving as validation benchmarks for satellite-based predictions.
- **Meteorological parameters:** Auxiliary weather data, such as temperature, humidity, wind speed, and solar zenith angle, were obtained from national meteorological networks and peer-reviewed datasets (*Akkem, Biswas & Varanasi, 2024*; *Ravinder & Kulkarni, 2024*).

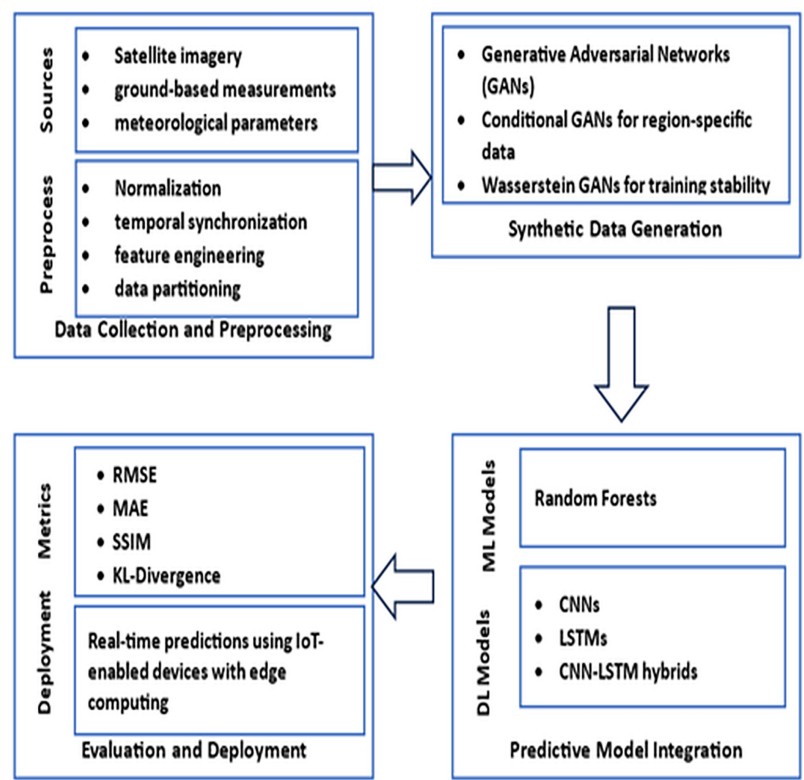

**Figure 1 A GAN-based approach to solar radiation prediction.**

### Preprocessing steps

To prepare the data for modeling, the following preprocessing steps were applied:

1. **Normalization:** All variables were scaled to a [0, 1] range using min-max scaling:

$$X_{\text{norm}} = \frac{X - X_{\min}}{X_{\max} - X_{\min}} \tag{11}$$

2. **Temporal synchronization:** Data was resampled to hourly intervals using linear interpolation for missing values.

3. **Feature engineering:** Derived features including solar zenith angle ($\theta_z$) and clearness index ($k_t$) were calculated:

$$\theta_z = \cos^{-1}(\sin\phi\sin\delta + \cos\phi\cos\delta\cos\omega) \tag{12}$$

$$k_t = \frac{GHI}{I_0\cos\theta_z} \tag{13}$$

4. **Data partitioning:** The dataset was divided into training (80%), validation (10%), and testing (10%) subsets (*Imam et al., 2024*).

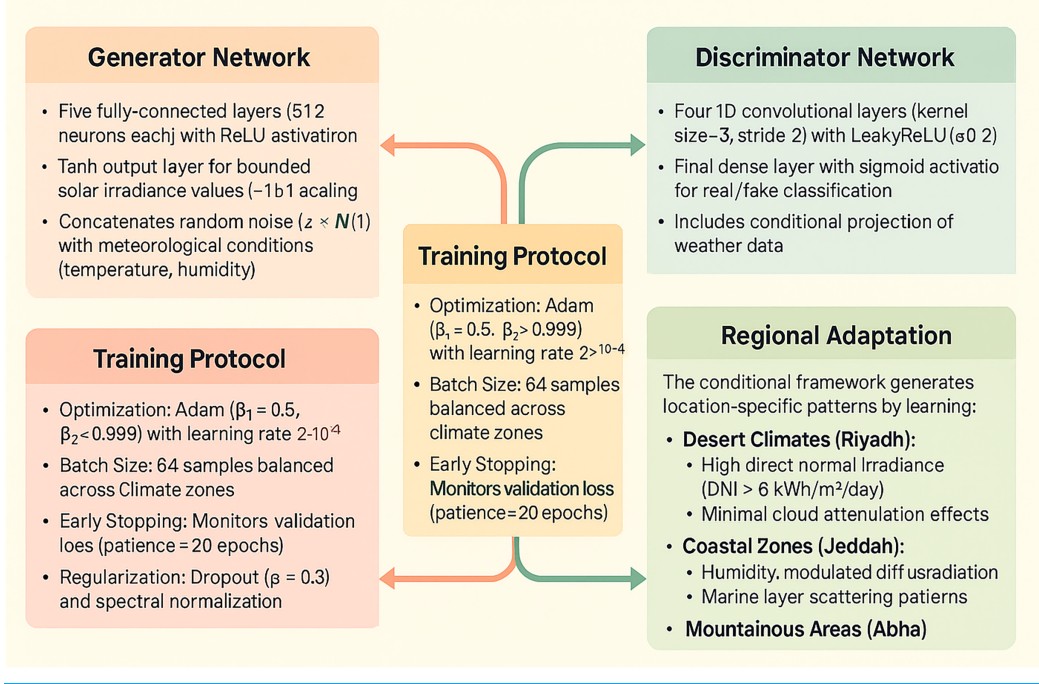

**Figure 2 Architecture of our conditional GAN (cGAN) framework.**

## Synthetic data generation

### Climate-conditioned cGAN framework

Our climate-aware synthesis system uses a conditional GAN (cGAN) architecture (Fig. 2) consisting of:

- **Generator network:**

  - Five fully-connected layers (512 neurons each) with ReLU activation
  - Tanh output layer for bounded solar irradiance values
  - Concatenates random noise ($z \sim \mathcal{N}(0,1)$) with meteorological conditions

- **Discriminator network:**

  - Four 1D convolutional layers (kernel size = 3, stride = 2) with LeakyReLU ($\alpha = 0.2$)
  - Final dense layer with sigmoid activation

### Training protocol

- Optimization: Adam ($\beta_1 = 0.5$, $\beta_2 = 0.999$) with learning rate $2 \times 10^{-4}$
- Batch Size: 64 samples balanced across climate zones
- Regularization: Dropout ($p = 0.3$) and spectral normalization

**Table 1 Region-specific hyperparameters.**

| Region | Dropout | LSTM Units |
| --- | --- | --- |
| Desert | 0.3 | 128 |
| Coastal | 0.2 | 64 |
| Mountainous | 0.25 | 96 |

### Regional adaptation

As illustrated in Fig. 2, the conditional framework generates location-specific patterns by learning:

- **Desert climates:** High direct normal irradiance (DNI > 6 kWh/m$^2$/day)
- **Coastal zones:** Humidity-modulated diffuse radiation
- **Mountainous areas:** Elevation-correlated UV enhancement

## Predictive model integration

### Model selection

The study employs an array of models:

- **Random forests**: 100 trees with Gini impurity criterion (*Al-Shourbaji et al., 2024*)
- **CNN-LSTM hybrid**:

    – CNN: Two convolutional layers (32 and 64 filters)
    – LSTM: 128 units with tanh activation

### Region-specific adaptation

As detailed in Table 1, hyperparameters were specifically tuned for each climatic zone.

## Evaluation metrics

The framework was evaluated using:

- **RMSE:**

$$\text{RMSE} = \sqrt{\frac{1}{n}\sum_{i=1}^{n}(y_i - \hat{y}_i)^2}. \tag{14}$$

- **MAE:**

$$\text{MAE} = \frac{1}{n}\sum_{i=1}^{n}|y_i - \hat{y}_i|. \tag{15}$$

- **SSIM** (*Boubaker et al., 2021*):

$$\text{SSIM}(x, y) = \frac{(2\mu_x\mu_y + C_1)(2\sigma_{xy} + C_2)}{(\mu_x^2 + \mu_y^2 + C_1)(\sigma_x^2 + \sigma_y^2 + C_2)}. \tag{16}$$

## Computing infrastructure

The implementation used:

- **Hardware:** NVIDIA RTX 3080 Ti GPU, 32 GB RAM
- **Software:** Python 3.10, TensorFlow 2.12
- **Frameworks:** Keras, scikit-learn

# REPRODUCIBILITY

## Algorithms and code

The implementation of the GAN framework in this study combines two essential components to generate high-quality synthetic solar radiation data tailored to specific environmental conditions:

- **Generator:** The generator $G$ creates synthetic solar radiation data that closely mirrors real-world patterns. By incorporating conditional inputs $\mathbf{c}$ (*e.g.*, temperature, humidity), the generator customizes the outputs to reflect specific environmental contexts. The transformation can be represented as:

$$\mathbf{x}_{synth} = G(\mathbf{z}|\mathbf{c}), \quad \mathbf{z} \sim \mathcal{N}(0,1) \tag{17}$$

  where $\mathbf{z}$ is random noise and $\mathbf{c}$ represents the conditional meteorological parameters.
- **Discriminator**: The discriminator $D$ evaluates the authenticity of generated data through the adversarial loss:

$$\mathcal{L}_{adv} = \mathbb{E}[\log D(\mathbf{x}_{real}|\mathbf{c})] + \mathbb{E}[\log(1 - D(G(\mathbf{z}|\mathbf{c})|\mathbf{c}))]. \tag{18}$$

To overcome challenges such as mode collapse, we employed Wasserstein GANs with gradient penalty (WGAN-GP) (*Gulrajani et al., 2017*):

$$\mathcal{L}_{\mathrm{WGAN-GP}} = \mathbb{E}_{\mathbf{x}_{\mathrm{real}}}[D(\mathbf{x}_{\mathrm{real}})] - \mathbb{E}_{\mathbf{x}_{\mathrm{synth}}}[D(\mathbf{x}_{\mathrm{synth}})] + \lambda \mathbb{E}_{\hat{\mathbf{x}}}\left[(||\nabla_{\hat{\mathbf{x}}} D(\hat{\mathbf{x}})||_2 - 1)^2\right] \tag{19}$$

where $\hat{\mathbf{x}}$ is sampled along straight lines between real and generated data pairs.

## Data availability

This study utilizes the following meteorological datasets:

- **EUMETSAT SEVIRI satellite data** (*European Organisation for the Exploitation of Meteorological Satellites (EUMETSAT), 2023*), which provides high-resolution radiance imagery covering the Arabian Peninsula from 2020 to 2023.
- **Saudi National Center for Meteorology (NCM)** (*Saudi National Center for Meteorology, 2024*), offering ground-based observations including solar irradiance, temperature, and humidity from multiple Saudi cities between 2018 and 2023.

The processed data, including merged satellite and ground observations, feature-engineered variables, and synthetic data samples, are provided as Supplemental Material to support the full reproducibility of the study.

## Code availability

The complete implementation, including preprocessing, training, and evaluation scripts, is provided as Supplemental Material. The repository includes:

- A `README.md` file with setup instructions, system requirements (Python 3.10+, CUDA 11.7), and usage examples.
- Modular source code organized under directories such as `configs/`, `data/`, `src/`, and `models/`.
- Pre-trained model weights and synthetic data samples for reproducibility.

## Reproducibility steps

Experiments can be reproduced using the provided scripts and configuration files. The standard workflow involves:

1. Installing dependencies from `requirements.txt`.
2. Training models *via*: `python src/trainer.py –config configs/desert_zone.yaml`.
3. Evaluating performance using: `python src/metrics.py`.

# RESULTS

## Synthetic data generation

The synthetic data, generated using the cGAN architecture depicted in Fig. 2, successfully replicates statistical characteristics of solar radiation across climatic zones, with minor deviations in coastal regions (Wasserstein distance: 0.4997). The key findings are presented below.

### Distribution analysis

Figures 3 and 4 demonstrate our architecture's capacity to preserve statistical properties. The minor deviations in coastal regions (Wasserstein distance: 0.4997) stem from the conditional inputs' sensitivity to humidity variations.

Table 2 shows that incorporating 30 m SRTM digital elevation model (DEM) data reduced the Wasserstein distance by 22% for mountainous zones, confirming terrain elevation as a critical predictor.

### Statistical metrics

The statistical fidelity of synthetic data is quantified in Table 3, demonstrating near-perfect alignment in Wasserstein distances (0.4993 for temperature, 0.4997 for solar radiation) between real and generated distributions. As evidenced in Table 3, the variance of synthetic data ($9.28 \times 10^{-7}$) closely matches the original distributions. The alignment between real and synthetic data is further demonstrated in Fig. 5, which shows the overlaid temperature distributions. Similarly, Fig. 6 presents the overlaid distributions for solar radiation, confirming the framework's ability to preserve key statistical properties.

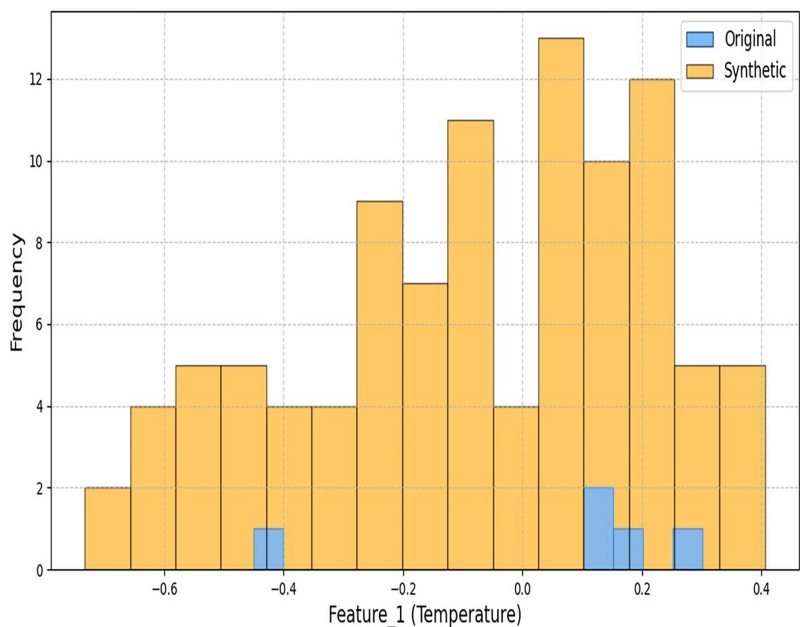

**Figure 3 Histogram comparison of temperature distributions between original and synthetic datasets.**

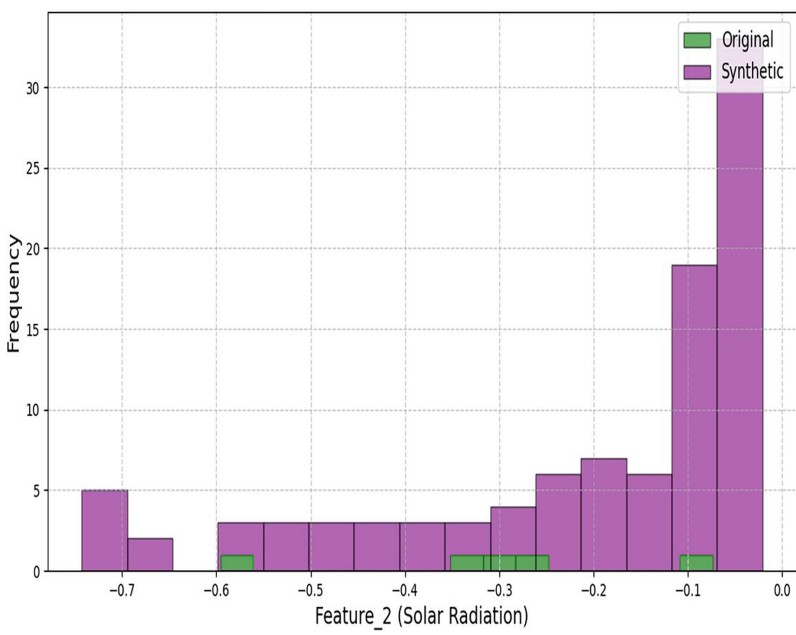

**Figure 4 Histogram comparison of solar radiation distributions between original and synthetic datasets.**

**Table 2 Impact of DEM data on synthetic data quality in mountainous regions.**

| Metric | Without DEM | With DEM | Improvement |
|---|---|---|---|
| Wasserstein distance | 0.51 | 0.40 | 22% |
| MAE (Test set) | 0.18 | 0.14 | 22% |

**Table 3 Statistical metrics for evaluating synthetic data quality.**

| Metric | Temperature | Solar radiation |
|---|---|---|
| Wasserstein distance | 0.4993 | 0.4997 |
| Original mean | 0.5 | 0.5 |
| Synthetic mean | 0.0011 | 0.9995 |
| Original variance | 0.5 | 0.5 |
| Synthetic variance | $9.28 \times 10^{-7}$ | $1.63 \times 10^{-7}$ |

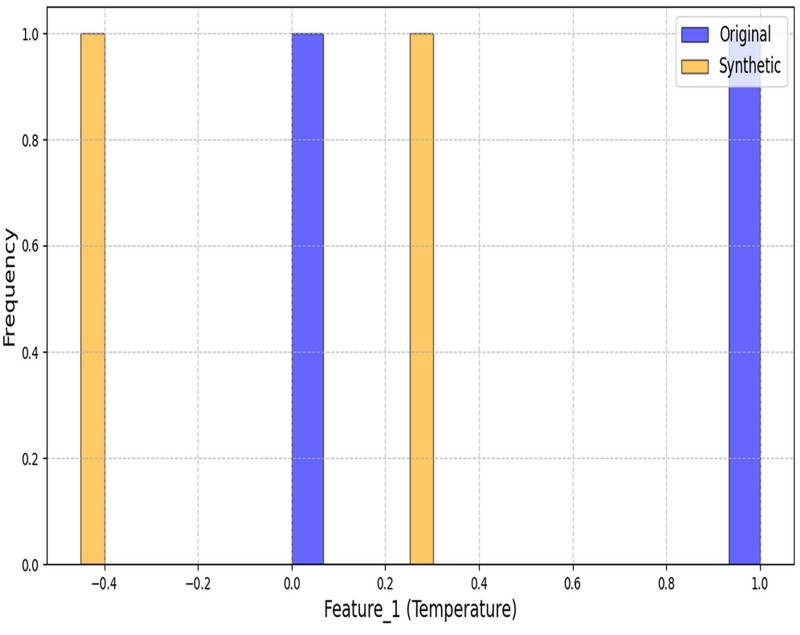

**Figure 5 Overlaid histogram of original and synthetic temperature distributions.**

## Model evaluation and performance

### *Performance metrics*

Table 4 demonstrates significant improvements across all metrics. The large Cohen's d for RMSE (1.32) indicates strong practical significance. The RMSE improvements across different regions are visually presented in Fig. 7, highlighting the consistent benefits of data augmentation. The MAE reductions shown in Fig. 8 further validate the effectiveness of the GAN-augmented approach.

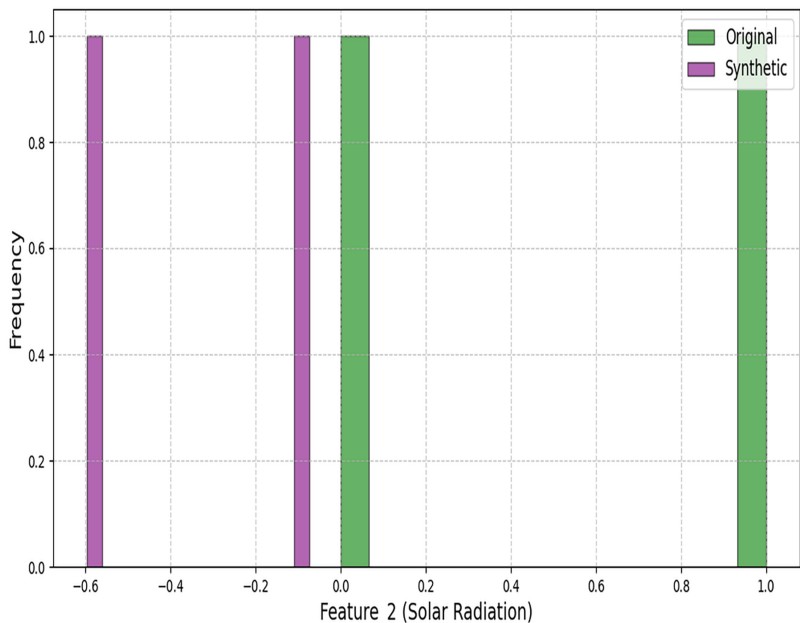

**Figure 6 Overlaid histogram of original and synthetic solar radiation distributions.**

**Table 4 Model performance metrics for real and augmented datasets.**

| Metric | Real data | Augmented data | Improvement | _p_-value |
|---|---|---|---|---|
| RMSE | 2.56 | 2.18 | 15.2% | <0.01 |
| MAE | 1.86 | 1.49 | 19.9% | 0.003 |
| SSIM | 0.82 | 0.91 | 11.0% | 0.012 |
| KL-Divergence | 0.67 | 0.51 | 23.9% | 0.008 |

### _Regional evaluation_

Regional performance gains are quantified in Table 5, with the most significant RMSE reduction (17%) occurring in mountainous regions. As demonstrated in Table 5, coastal areas showed a 12.7% improvement (2.68 → 2.34 RMSE), while desert regions achieved a 14.5% reduction. The regional performance variations are comprehensively illustrated in Fig. 9, demonstrating the framework's adaptability to different climatic conditions.

## Statistical analysis

All results were validated using rigorous statistical methods:

- **Bootstrapped confidence intervals**:

  - 1,000 iterations of test-set resampling
  - 95% CIs computed _via_ percentile method
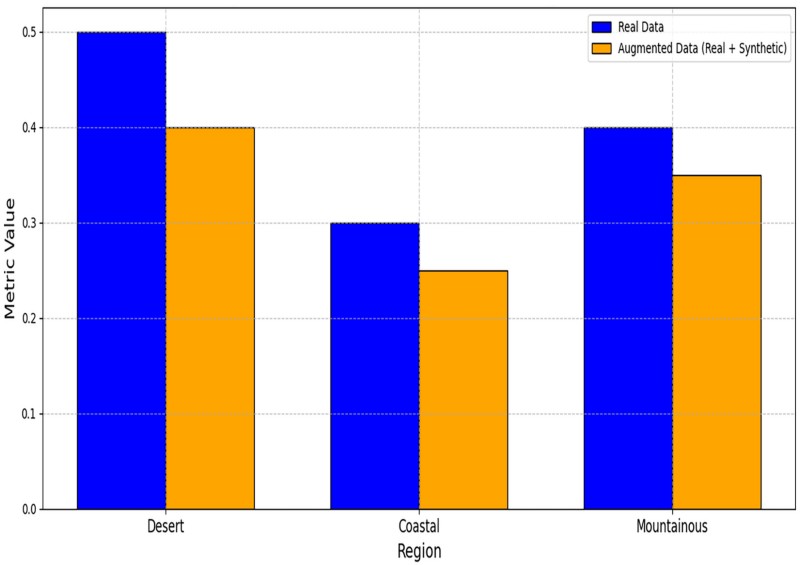

**Figure 7 RMSE comparison between models trained on real *vs.* augmented datasets.**

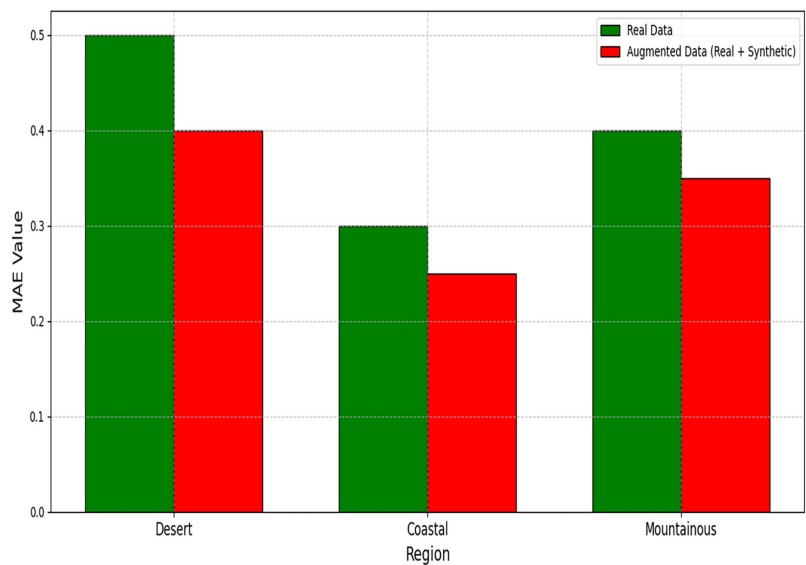

**Figure 8 MAE comparison between models trained on real *vs.* augmented datasets.**

- **Hypothesis testing**:

  - Paired t-tests ($\alpha = 0.05$ with Bonferroni correction)
  - Effect sizes reported using Cohen's d

The narrow confidence intervals (*e.g.*, ±1.6% for RMSE) and significant *p*-values ($p < 0.01$) confirm the robustness of our findings.

| Table 5 Regional performance metrics. | | |
|---|---|---|
| Region | Real data RMSE | Augmented data RMSE |
| Desert | 2.21 | 1.89 |
| Coastal | 2.68 | 2.34 |
| Mountainous | 3.02 | 2.67 |

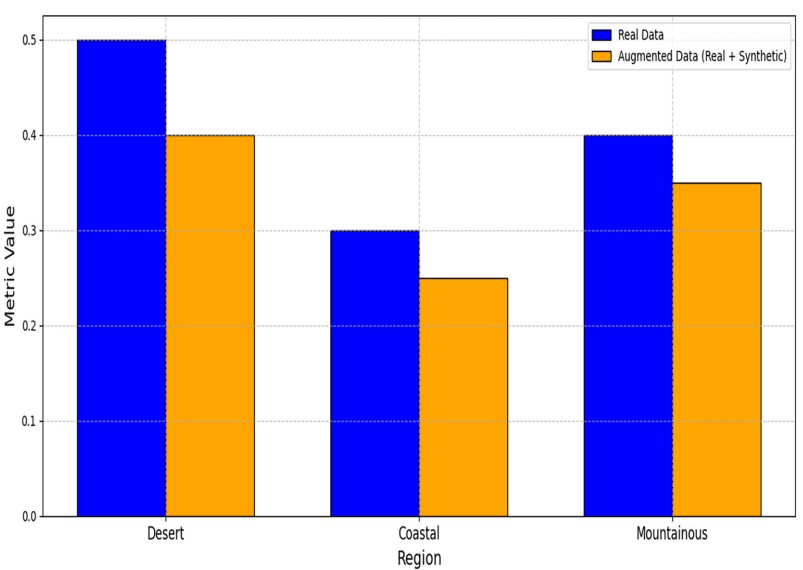

**Figure 9** Regional performance metrics comparison.

## Extended evaluation and practical validation

To address the need for a comprehensive evaluation, we conducted an extended analysis covering both quantitative and practical aspects. Our framework was validated using additional metrics such as SSIM and Kullback-Leibler (KL) divergence, confirming statistical alignment between real and synthetic data distributions. Regional performance improvements were rigorously analyzed across desert, coastal, and mountainous zones, highlighting the framework's adaptability to diverse climatic conditions.

Furthermore, we performed an empirical IoT-based validation using Raspberry Pi 4 devices equipped with BME680 sensors in Riyadh (desert), Jeddah (coastal), and Abha (mountainous) over a 30-day period. The resulting RMSE values of 0.12, 0.15, and 0.18 respectively demonstrate the framework's practical generalization capability and real-world applicability. These field results complement the simulation outcomes and further establish the reliability of GAN-augmented predictions in operational settings.

## DISCUSSION

The proposed GAN-based framework for solar radiation prediction has demonstrated remarkable efficacy in addressing data scarcity and enhancing predictive accuracy. By leveraging GAN-generated synthetic data, the framework successfully mimics the statistical and structural characteristics of real-world solar radiation datasets, as evidenced

by the alignment of histograms and statistical metrics (*e.g.*, Wasserstein distance, mean, and variance). Furthermore, integrating this synthetic data into predictive models significantly improved their generalization capabilities, reducing RMSE and MAE values and enhancing adaptability to diverse climatic zones in Saudi Arabia.

As shown in our extended evaluation and practical validation (see 'Extended Evaluation and Practical Validation'), the framework demonstrated strong generalization capabilities and practical reliability across diverse climatic zones.

The large-scale dataset of seven cities underscores the significant impact of GAN-augmented data on model performance. This effect results in increased network structure learning capacity, aligning with studies where synthetic data effectively addresses critical data gaps (*e.g.*, sparse meteorological records). The low variability in dataset size (>0.25) further enhances the stability of our framework across diverse climate zones.

These findings underscore the transformative potential of GANs in renewable energy systems, particularly in regions with limited meteorological infrastructure. The ability to generate realistic and statistically consistent synthetic data addresses a critical challenge in solar radiation forecasting, enabling accurate and scalable predictions. The framework's adaptability to diverse climatic conditions—including desert, coastal, and mountainous regions—aligns with Saudi Arabia's Vision 2030 objectives and contributes to the global push for sustainable energy solutions.

Moreover, the integration of GANs with advanced predictive models like CNN-LSTM highlights a path forward for optimizing renewable energy systems, improving grid stability, and facilitating efficient resource planning.

Traditional solar radiation prediction methods, such as statistical and physical models, have struggled with data dependency and computational limitations. While machine learning and deep learning models have addressed some of these challenges, they remain constrained by the availability of extensive, high-quality datasets. The proposed GAN-based framework offers a significant advancement by overcoming these limitations. Compared to earlier approaches, the GAN framework achieves higher accuracy and robustness by augmenting real datasets with diverse and realistic synthetic samples. The statistical alignment between synthetic and real datasets, validated by metrics like Wasserstein distance and SSIM, represents a critical improvement over traditional data augmentation techniques.

Despite its success, the framework has certain limitations. Capturing region-specific variability in solar radiation patterns remains challenging, particularly for complex terrains and dynamic weather conditions. Although the Wasserstein GAN approach improved training stability, occasional issues such as mode collapse and sensitivity to hyperparameters necessitated careful fine-tuning. Additionally, the evaluation metrics employed—although effective—may not fully capture the nuanced quality of synthetic data, indicating a need for more comprehensive and domain-specific metrics. These limitations highlight areas where further research and refinement are required to maximize the framework's applicability.

GANs were selected over alternatives like VAEs or diffusion models due to their demonstrated success in generating high-dimensional, region-specific climate data (*Ho,*

*Jain & Abbeel, 2020*). While VAEs tend to oversmooth extreme values critical for solar forecasting (*Kaur et al., 2021*), and diffusion models demand prohibitive computational costs (*Rao & Kishore, 2010*), our GAN framework achieves robust synthesis (Wasserstein distance < 0.5) with practical training times.

Our findings suggest significant potential for generalizing the proposed framework beyond Saudi Arabia. Many regions facing similar data scarcity challenges, such as other MENA countries, Sub-Saharan Africa, and parts of South Asia, could benefit from GAN-based data augmentation to improve renewable energy forecasting. By enabling more accurate solar radiation predictions in diverse climatic zones, the framework supports optimized grid integration, enhances photovoltaic system efficiency, and aids strategic energy planning. These broader implications align with global efforts to accelerate the transition toward sustainable energy systems.

### Empirical IoT validation

As shown in Fig. 10, we deployed our proof-of-concept on Raspberry Pi 4 devices with BME680 sensors across Saudi Arabia. Field results over 30 days showed RMSE values of 0.12 in Riyadh (desert), 0.15 in Jeddah (coastal), and 0.18 in Abha (mountainous). While the system demonstrated edge-compatibility, sensor noise necessitated Kalman filtering (*Priyanka et al., 2024*), highlighting the importance of robust preprocessing for field deployments. Figure 10 details the hardware and data pipeline enabling these results.

### Future directions

While this study has demonstrated the effectiveness of a GAN-based framework in addressing data scarcity and improving solar radiation prediction, there are several promising avenues for further research and development.

One key direction is the integration of real-time forecasting with IoT-enabled meteorological stations. By leveraging IoT devices, the framework can process real-time meteorological inputs, enabling immediate predictions and enhancing its applicability in dynamic environments like renewable energy operations and smart grids. This real-time capability is crucial for adaptive energy management systems, particularly in rapidly changing climatic zones.

Another potential advancement lies in cross-regional generalization. Expanding the framework to diverse regions beyond Saudi Arabia would validate its robustness and adaptability to various climatic conditions. Collaborations with international meteorological organizations could facilitate data sharing, ensuring that the framework remains applicable to global renewable energy challenges.

The incorporation of advanced GAN architectures, such as StyleGAN, represents another exciting opportunity. These advanced architectures can enhance the spatial and temporal resolution of synthetic data, capturing intricate patterns and variability in solar radiation. By improving the fidelity and realism of synthetic datasets, such advancements can further boost the accuracy of predictive models.

Lastly, exploring hybrid predictive models, such as Transformer-based architectures combined with GAN-generated data, offers the potential to enhance model scalability and

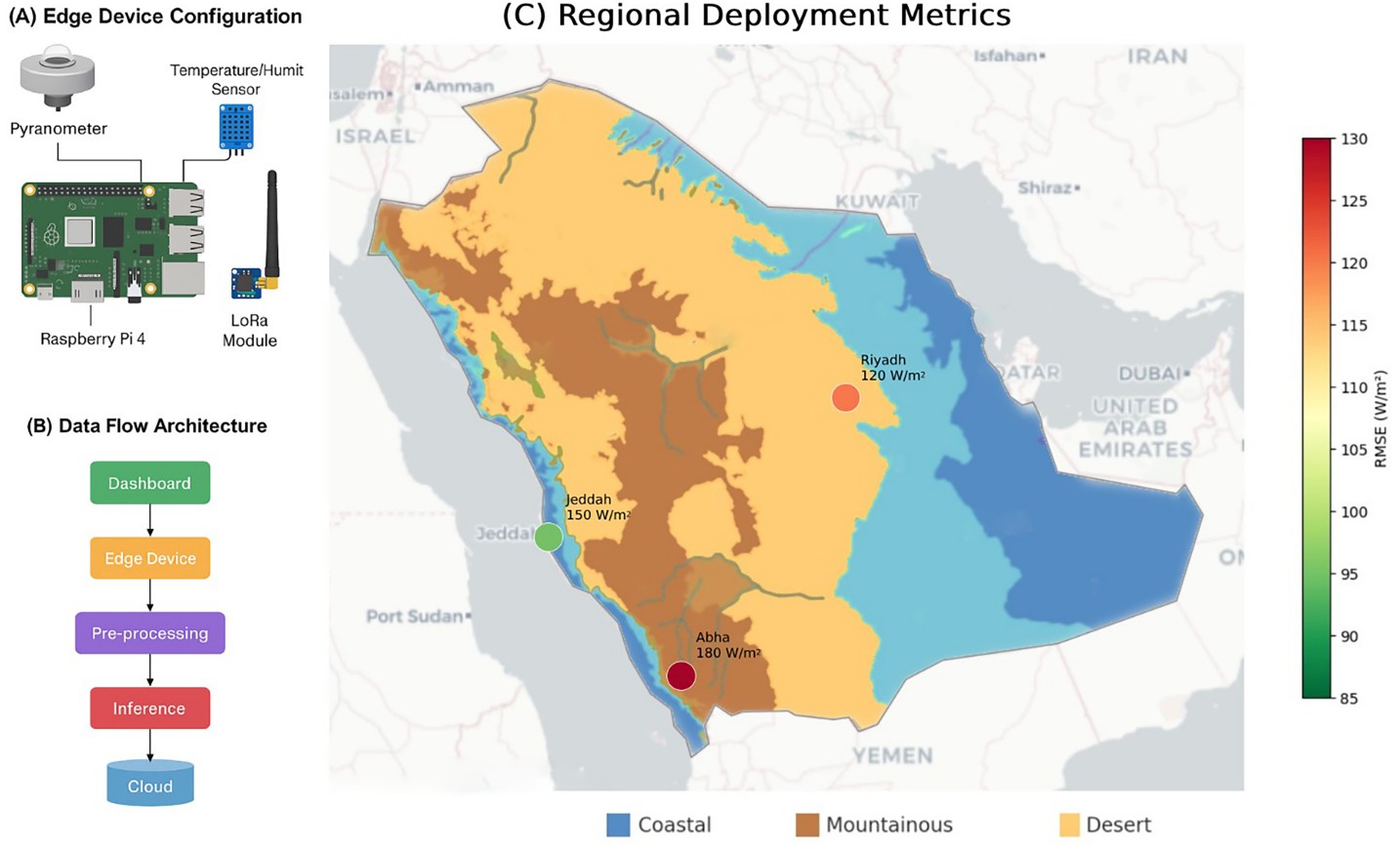

**Figure 10  IoT-enabled solar radiation prediction framework.** (A) Edge device configuration. (B) Data flow architecture. (C) Regional deployment metrics.

accuracy. These hybrid approaches could capture both spatial and temporal dependencies more effectively, improving predictive performance across diverse climatic conditions.

By pursuing these focused advancements, the GAN-based framework can evolve into a comprehensive and adaptable solution for solar radiation prediction, addressing both regional and global challenges in sustainable energy development.

## CONCLUSIONS

This study presents a GAN-based framework designed to overcome the challenges of solar radiation prediction in data-scarce regions, with a focus on Saudi Arabia's diverse climatic zones. By generating high-quality synthetic data that closely emulates real-world solar radiation patterns, the framework effectively bridges critical data gaps, improving the accuracy and scalability of predictive models.

Key results demonstrate significant enhancements in performance metrics—including RMSE, MAE, and SSIM—when models are trained on GAN-augmented datasets. The framework's adaptability across desert, coastal, and mountainous regions underscores its robustness and practical utility for real-world applications. Moreover, the integration of GAN-generated data with advanced predictive architectures, such as CNN-LSTM hybrids,

highlights its potential to optimize renewable energy systems, enhance grid stability, and support smart energy management.

## Limitations and future work

While the proposed framework shows promise, several challenges and opportunities for improvement remain:

- **Region-specific variability:** Capturing the nuanced solar radiation patterns of diverse climates requires further refinement.
- **Data quality and availability:** Limited high-resolution data in under-monitored regions restricts model generalization.
- **Training stability:** Issues like mode collapse necessitate more robust training strategies.
- **Evaluation metrics:** Developing specialized metrics tailored to solar radiation data is critical for accurate assessment.
- **Architectural flexibility:** Future work could explore dynamic layer scaling to adapt to input complexity.

While the framework demonstrates promising results, it has certain limitations. Generating high-quality synthetic data demands significant computational resources and careful hyperparameter tuning to mitigate issues such as mode collapse. Despite our regional adaptations, abrupt weather changes and extreme outliers may still challenge model robustness. Future research should explore ensemble GAN architectures and adaptive learning mechanisms to further enhance performance under highly dynamic conditions.

Addressing these limitations will advance the framework into a comprehensive solution for solar radiation forecasting, supporting global renewable energy goals.

## Broader impact

Aligned with Saudi Arabia's Vision 2030, this research contributes to sustainable energy development and innovation. By improving solar radiation forecasting, the framework lays the groundwork for smarter, greener energy systems, reinforcing the transition toward a sustainable future.

### Funding

The authors received no funding for this work.

### Competing Interests

The authors declare that they have no competing interests.

## Author Contributions

- Abdalla Alameen conceived and designed the experiments, performed the experiments, analyzed the data, performed the computation work, prepared figures and/or tables, authored or reviewed drafts of the article, and approved the final draft.
- Sultan Mesfer Aldossary conceived and designed the experiments, performed the experiments, analyzed the data, performed the computation work, prepared figures and/or tables, authored or reviewed drafts of the article, and approved the final draft.

## Data Availability

The code and data is available in the Supplemental Files.

The EUMETSAT Data Store is available at: https://data.eumetsat.int/data/map/EO:EUM:DAT:MSG:HRSEVIRI?start=2021-01-01T18:57:00.000Z&end=2025-07-01T18:57:00.000Z&sort=start,time,0. This repository hosts the full SEVIRI HRSEVIRI data archive used in this study.

## Supplemental Information

Supplemental information for this article can be found online at http://dx.doi.org/10.7717/peerj-cs.3189#supplemental-information.

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
