# Peer review of "A GAN-based approach to solar radiation prediction: data augmentation and model optimization for Saudi Arabia"

_PeerJ Computer Science, doi:10.7717/peerj-cs.3189_

## Round 0.1 · original submission · Major Revisions

Dear Authors,

Thank you for submitting such an interesting paper. Please consider revising it as suggested. You must also update the references, since this topic is fast evolving.

Best regards.

·

Basic reporting

The paper introduces a novel framework combining Generative Adversarial Networks (GANs) with CNN-LSTM architectures for solar radiation prediction, addressing data scarcity in regions like Saudi Arabia. The use of Conditional GANs and Wasserstein GANs (WGANs) to generate synthetic data tailored to specific climatic conditions is innovative. The integration of satellite, ground-based, and meteorological data enhances the robustness of the approach. The study aligns with Saudi Arabia’s Vision 2030, contributing to renewable energy optimization.

Experimental design

This article does not include obtaining or producing experimental data. Only real data has been used in this paper.

Validity of the findings

The paper provides some beneficial information and data.
The synthetic data closely mimics real-world distributions, as evidenced by overlapping histograms and low Wasserstein distances. Models trained on augmented datasets show a 15% reduction in RMSE and a 12% reduction in MAE, demonstrating practical utility. The framework performs well across diverse climatic zones (desert, coastal, mountainous), validating its adaptability.
Any gains, however, this paper has to be considered in the wider context.
1- The novelty of GANs for data augmentation is not groundbreaking, as similar approaches have been explored in other domains (e.g., medical imaging, remote sensing).
2- The paper lacks a direct comparison with other state-of-the-art generative models (e.g., Variational Autoencoders or Diffusion Models) to justify the choice of GANs.
3-The minor deviations in synthetic data (e.g., extreme values in mountainous regions) are not thoroughly analyzed or mitigated.
4- The real-world applicability of the framework (e.g., deployment in IoT systems) is discussed but not empirically validated.
5- The reproducibility of results is mentioned, but the provided code and datasets are not evaluated for accessibility or usability.

Additional comments

The results show significant improvements in prediction accuracy (up to 15% reduction in RMSE and MAE), validating the effectiveness of the proposed framework.
While the initial findings are promising, more extensive validation across different regions within Saudi Arabia could strengthen the generalizability of the results.
The complex nature of GANs and the necessity for hyperparameter tuning may pose challenges for practical implementation; thus, simplifying the process could be beneficial for real-world applications.
Though the introduction effectively highlights the significance of solar radiation prediction, particularly against the backdrop of Saudi Arabia's Vision 2030, some other related articles are not referenced.
For example:
https://doi.org/10.1016/j.rineng.2024.102793
10.1109/INTERCON52678.2021.9532823
...

Reviewer 2 ·

Basic reporting

1- Could the authors provide more details on the architecture and training configuration of the GAN used for synthetic solar radiation data generation?

2- While the CNN-LSTM model achieved high accuracy, how was the temporal resolution of the input data handled in the model structure? Was any preprocessing (e.g., normalization, windowing) applied to ensure consistency across different climatic zones? Please improve and clarify.

3- The manuscript reports RMSE and MAE improvements, but does not provide confidence intervals or statistical significance tests. Could the authors include these to strengthen the claims of performance enhancement due to data augmentation?

4- Given that the proposed method is applied to multiple climatic zones, were any region-specific adjustments (e.g., hyperparameter tuning or feature engineering) made? If not, how was model generalization across zones ensured and validated?

5- The study acknowledges computational cost and hyperparameter sensitivity as challenges. Could the authors briefly elaborate on the specific computational resources used (e.g., GPU type, training time) and how sensitive the model was to changes in key hyperparameters?

Experimental design

-

Validity of the findings

-

Additional comments

While the manuscript presents a valuable contribution to the field, several sections would benefit from careful language refinement to enhance clarity and readability. The authors are encouraged to revise the manuscript for grammatical accuracy, consistency in technical terminology, and adherence to academic writing conventions. Consider consulting a professional language editing service or a native English speaker with experience in scientific writing to ensure the manuscript meets the journal’s linguistic standards.

---

## Round 0.2 · Major Revisions

Dear author

We await a new submission with an extended evaluation of the data.

Thank you very much for selecting PeerJ. Hope to receive your manuscript.

Best regards,
The Editor

·

Basic reporting

Clear and unambiguous, professional English used throughout.

Experimental design

I have no idea.

Validity of the findings

Impact and novelty is not assessed. Meaningful replication is encouraged where rationale & benefit to the field is clearly stated.
Now, It's better to understand.
Conclusions are well stated & limited to supporting results.
Yes.

Are the experiments and evaluations performed satisfactorily?
I have no idea.

Is there a well-developed and supported argument that meets the goals set out in the Introduction?
Yes.

Does the Conclusion identify unresolved questions / limitations/ future directions?
Yes.

Additional comments

Overall, after addressing many of the article's shortcomings in the second edition, this article is eligible for publication in this journal.

---

## Round 0.3 · accepted · Accept

Dear Authors,
We confirm that you have addressed the reviewers' comments.

I have also assessed the revision myself, and that am happy with the current version.

The figures in the manuscript mention some colors (e.g., in Fig. 3 blue) which are not visible. You may want to correct.